# An in vivo gene amplification system for high level expression in *Saccharomyces cerevisiae*

Bingyin Peng [1,2,3,4✉], Lygie Esquirol [1,5], Zeyu Lu[1,3,4], Qianyi Shen[1,3,4], Li Chen Cheah [1,3], Christopher B. Howard[1], Colin Scott [2,6], Matt Trau [1,7], Geoff Dumsday[8] & Claudia E. Vickers [2,3,4,5✉]

Bottlenecks in metabolic pathways due to insufficient gene expression levels remain a significant problem for industrial bioproduction using microbial cell factories. Increasing gene dosage can overcome these bottlenecks, but current approaches suffer from numerous drawbacks. Here, we describe HapAmp, a method that uses haploinsufficiency as evolutionary force to drive in vivo gene amplification. HapAmp enables efficient, titratable, and stable integration of heterologous gene copies, delivering up to 47 copies onto the yeast genome. The method is exemplified in metabolic engineering to significantly improve production of the sesquiterpene nerolidol, the monoterpene limonene, and the tetraterpene lycopene. Limonene titre is improved by 20-fold in a single engineering step, delivering ~1 g L$^{-1}$ in the flask cultivation. We also show a significant increase in heterologous protein production in yeast. HapAmp is an efficient approach to unlock metabolic bottlenecks rapidly for development of microbial cell factories.

[1] Australian Institute for Bioengineering and Nanotechnology (AIBN), The University of Queensland, Brisbane, QLD 4072, Australia. [2] CSIRO Synthetic Biology Future Science Platform, Commonwealth Scientific and Industrial Research Organisation (CSIRO), Black Mountain, ACT 2601, Australia. [3] ARC Centre of Excellence in Synthetic Biology, Queensland University of Technology, Brisbane, QLD 4000, Australia. [4] Centre of Agriculture and the Bioeconomy, School of Biology and Environmental Science, Faculty of Science, Queensland University of Technology, Brisbane, QLD 4000, Australia. [5] Griffith Institute for Drug Discovery, Griffith University, Brisbane, QLD 4111, Australia. [6] Biocatalysis and Synthetic Biology Team, CSIRO Land and Water, Black Mountain Science and Innovation Park, Canberra, ACT 2061, Australia. [7] School of Chemistry and Molecular Biosciences (SCMB), The University of Queensland, Brisbane, QLD 4072, Australia. [8] CSIRO Manufacturing, Clayton, VIC 3169, Australia. ✉email: bingyin.peng@qut.edu.au; claudia.vickers@qut.edu.au

To achieve economically viable rates, yields and titres for a given product in microbial cell factories, it is commonly necessary to increase expression of introduced genetic constructs[1,2]. This is typically achieved by manipulating transcription levels via transcriptional control elements (promoters and other genetic sequences)[3]. However, this approach is subject to thresholds on individual constructs. This often means that expression levels are insufficient for a desired application. For example, enzymes with poor catalytic properties that cannot be improved by enzyme engineering represent significant flux bottlenecks in metabolic engineering[4]. In addition, where extremely high product levels are required (e.g., protein production systems), very high expression can deliver a direct economic benefit to the bioprocess. Increasing the gene dosage can be used to overcome transcriptional thresholds and increase expression levels.

The brewer's yeast *Saccharomyces cerevisiae* is a eukaryotic model organism and an important industrial microorganism for production of biofuels, biochemicals, and biopharmaceuticals. In *S. cerevisiae*, multi-copy yeast episomal plasmids or genome integration into ribosomal DNA (rDNA) sites are typically used to increase gene dosage[5–8]. However, these approaches are not stable in the absence of selection pressure, and plasmids can suffer from copy number instability leading to variable expression levels[5–8]. In addition, use of selection systems in industrial processes adds additional costs and often is not scalable[9,10]. To stabilise strains without the need for selective antibiotic or auxotrophy systems, auto-selection markers such as glycolytic genes (*FBA1*, fructose-bisphosphate aldolase; *POT1/TPI1*, triosephosphate isomerase) can be used[5,11,12]. However, this requires the background strains to have the correct genotype for knock-out. Transposable elements can also be used for multi-copy integration, however variable copies are integrated at random loci on genome, which means integrated components cannot be removed to facilitate future engineering steps (for example, swapping terpenoid synthases for different terpenoid production platforms)[13–17]. A method overcoming all these limitations is highly desirable.

Gene amplification commonly happens in nature during cell proliferation, as part of molecular evolution, as well as in some laboratory experiments[2,18–23]. In yeast, tandem amplification of fitness-associated genes on the genome permits improved survival and propagation of cells under new or changing conditions[18–20]. For example, amplification of the xylose isomerase, cellobiose-utilisation, and copper resistance (CUP1) genes occurs over prolonged adaptive cultivation on xylose[19,20], cellubiose[24], and copper ions[25], respectively. Another example is the amplification of tandem repeated rDNA under some conditions[26]. These examples demonstrate that if the expression level of a gene product is tightly linked to growth fitness and cannot meet the needs for maximum growth, gene amplification can occur through adaptive evolution.

In diploids, haploinsufficiency describes a state whereby one allele at a heterozygous locus provides little or no product, and the combined product from both alleles is insufficient to deliver the wild type phenotype[27]. Expression dosage of haploinsufficient genes links tightly with the growth fitness in yeast[28]. This can be explored as an evolutionary force to drive gene amplification and as a selection pressure for maintenance of the amplified constructs under normal cultivation conditions.

Here, we design an artificial genetic structure that enables amplification of a haploinsufficient gene through tuning of its promoter strength or translational efficiency (HapAmp). This structure is incorporated into genetic vectors which can be used to introduce multiple copies of linked heterogeneous genes on the genome. We exemplify the applications of this technique by

developing yeast factories for improved production of terpenes by metabolic engineering and for high production of pharmaceutically relevant proteins.

## Results

**Construct design for in vivo gene amplification.** Two elements are required for gene amplification to occur: (1) a gene linked to cell fitness, and (2) homologous DNA sequences to support recombination[20]. In addition, a strong replication origin can promote amplification[29–31]. These three elements exist in tandem repeat in the rDNA region and the *CUP1* region in the yeast genome (Fig. 1a).

We designed a genetic structure for gene amplification in yeast (Fig. 1b). The construct has recombination arms at each end. Arm 1 is homologous to the promoter region of a haploinsufficient gene, and Arm 2 is homologous to the initial part of the haploinsufficient gene open reading frame. This allows insertion of the construct into the genome by homologous recombination. Downstream of Arm 1 are a selectable marker for transformation selection and homologous Arm 3, which is homologous to the terminator region of the haploinsufficient gene. Between Arm 3 and Arm 2, there are an autonomous replicating sequence (ARS) and a promoter. The promoter is weaker than the native promoter of the haploinsufficient gene and positioned such that integration results in substitution of the native promoter of the haploinsufficient gene with the weaker

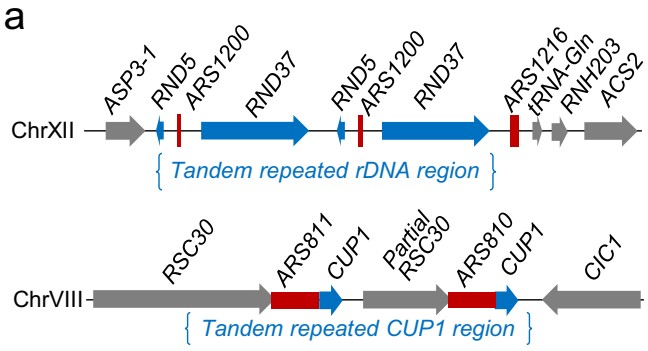

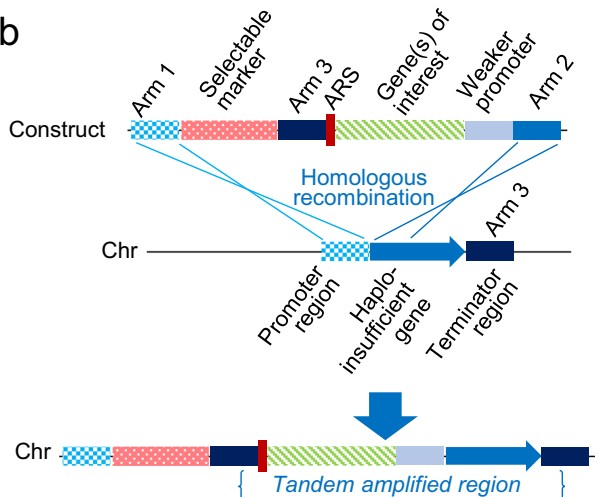

**Fig. 1 Design of in vivo gene amplification. a** Natural genome structures at the ribosomal DNA (rDNA) locus on chromosome XII and the *CUP1* locus on chromosome VII. **b** Construct design for in vivo gene amplification (HapAmp). ARS autonomous replicating sequence.

promoter. Genes of interest, to be expressed heterologously, can be inserted between Arm 3 and the weaker promoter.

Driving expression through a weaker promoter attenuates the protein yield from each copy of the haploinsufficient gene. This, in turn, is expected to decrease the growth rate in yeast. Native amplification of the region between homologous Arm 3 will then occur as yeast evolves towards faster growth.

**Using *RPL25* or *SEC23* haploinsufficient gene loci to drive amplification.** The effect of haploinsufficient genes on growth fitness has been characterised previously[28]. We used the ribosomal 60S subunit protein L25 (*RPL25*) and the *SEC23*-encoding component of the Sec23p-Sec24p heterodimer of the COPII vesicle coat. These two genes have the strongest fitness effect in rich medium and in minimal mineral medium[28]. We developed four constructs with *RPL25* as the driving gene, *LEU2* as selection marker, and an early-firing ARS *ARS306*[32] to facilitate amplification; and three constructs with *SEC23* as the driving gene, hygromycin B resistant gene *hphMX* as selection marker, and the strong *ARS1max* ARS[33] to facilitate amplification (Fig. 2a).

To identify promoters with suitable expression strengths, promoters were selected from the wide variety of promoters we previously analysed[34], to test with each target locus (Fig. 2a, d).

For the *RPL25* constructs we used the *YEF3* promoter (which has similar strength to the *RPL25* promoter; Construct 1) and the *ERG1*, *PDA1*, or *BTS1* promoters (all with multiple-fold weaker expression than *RPL25* promoter; Constructs 2–4). For the *SEC23* constructs, we used the *ERG1* promoter (stronger than the *SEC23* promoter; Construct 5), the *GLO2* promoter, or the *COG7* promoter (both multiple-fold weaker than the *SEC23* promoter; Constructs 6 and 7). An eighth promoter construct was designed and tested later (see below). We used yeast-enhanced green fluorescent protein (yEGFP) under the control of the *TEF1* promoter and the *URA3* terminator as the gene of interest and as a reporter for proof of concept.

The seven constructs were transformed into *S. cerevisiae* CEN.PK strains. Transformation plates were screened by imaging yEGFP fluorescence under blue light (Supplementary Fig. 1a, c) and colonies were selected for increased fluorescence. For each construct, six strongly fluorescing clones were selected. Visual observation after sub-culturing demonstrated an inverse correlation between promoter strength (Fig. 2d) and GFP fluorescence (Supplementary Fig. 1b). Three clones with similar fluorescence were selected for quantitative characterisation for each construct.

Where promoter strength was similar or greater than the native promoter, yEGFP was found at a single copy on the genome (Fig. 2c: Constructs 1 and 5), and fluorescence (Fig. 2e:

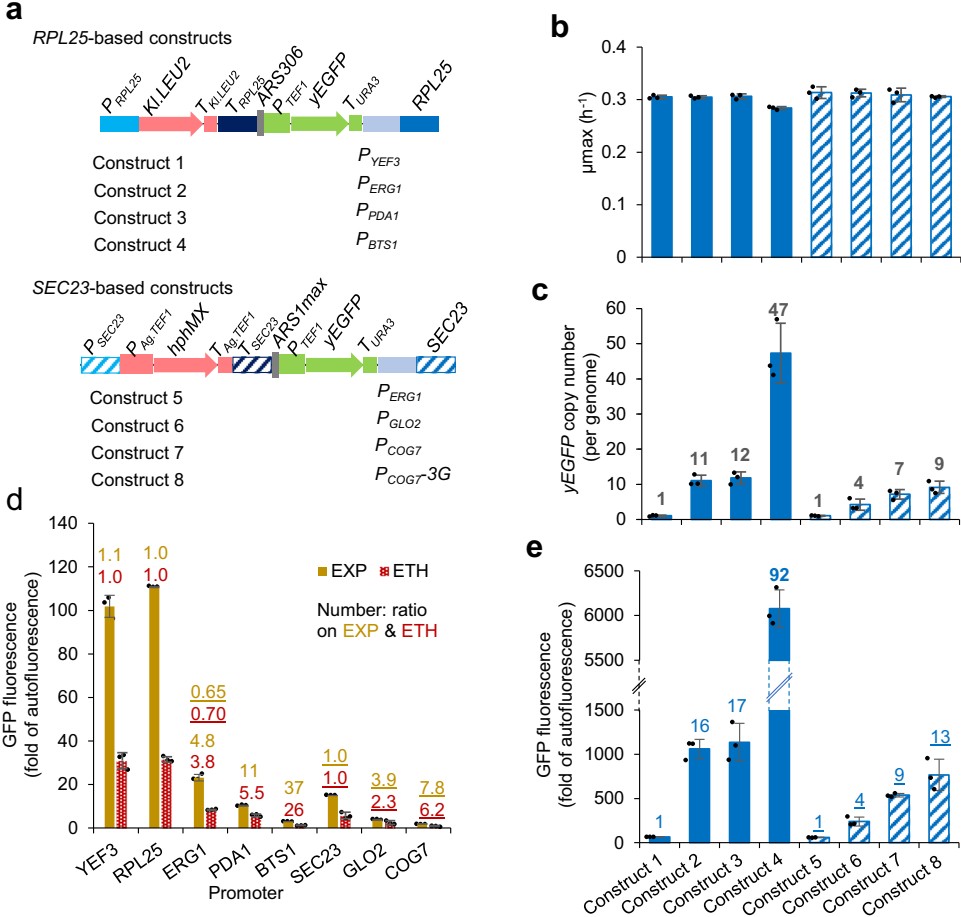

**Fig. 2 Design and characterisation of gene amplification constructs for haploinsufficient target genes *RPL25* or *SEC23*. a** Schematic of gene amplification constructs. **b**, **c**, **e** Maximum growth rate, yEGFP (yeast-enhanced green fluorescent protein) gene copy number, and yEGFP fluorescence in strains transformed with the constructs in **a**. Strains were selected by brightness of yEGFP fluorescence (Supplementary Fig. 1). **d** Promoter characterisation using yEGFP as the reporter in the cells at the exponential growth phase (EXP) and the post-diauxic-shift growth phase (ETH) when ethanol was used as the carbon source. Yeast cells were grown in microplates in **d** and in flasks in **b**, **c**, **e**. yEGFP fluorescence is expressed as percentage of exponential-phase auto-fluorescence of the reference strain. The numbers were calculated by dividing the mean value for *RPL25* or *SEC23* (underlined) by the mean value. Mean values ± standard deviations are shown ($N = 3$ independent biological replicates). Source data are provided as a Source Data file.

Constructs 1 and 5) was similar to fluorescence we observed previously in strains with a single copy of the $P_{TEF1}$-$yEGFP$-$T_{URA3}$ construct[3]. yEGFP gene copy number and fluorescence both increased where the native promoter was substituted for weaker promoters (Fig. 2c, e: Constructs 2–4, 6, 7). Copy number increased from 4-fold to 47-fold, whereas fluorescence increase was 4-fold to 92-fold. There was a strong positive correlation between copy number and fluorescence ($r^2 = 0.985$), and a weak negative correlation between fluorescence and promoter strength/ copy number ($r^2 = 0.376$ and $0.694$ respectively). The most remarkable result was where the $RPL25$ promoter was substituted for the $BTS1$ promoter; this resulted in ~47 copies of yEGFP per genome and a ~92-fold increase yEGFP fluorescence (Fig. 2c, e).

To further increase copy number at the $SEC23$ locus, we attenuated translation by making a construct with three non-preferred glycerine codons (GGA) inserted following the start codon of $SEC23$ under the control of the $COG7$ promoter (Fig. 2a: Construct 8), which delivered the most gene amplification in the first round (9 copies). A slight increase in gene copy and fluorescence was obtained (Fig. 2c, e). Translational down-regulation by use of non-preferred codons provides a second mechanism to drive an increase in copy number for genes at haploinsufficient gene loci.

In the initial design (Fig. 1), we include ARS in the module basing on the genetic features at naturally amplified genomic loci. To confirm the role of ARS in the current system, we removed the ARS sequence in the Construct 3. The ARS-removed construct could lead to the formation of the very fluorescent colonies after transformation (Supplementary Fig. 1). This indicates that ARS may not be essential for HapAmp.

Increased copy number did not negatively impact the growth rate of any of the strains except for clones with the $P_{BTS1}$-$RPL25$ construct (Fig. 2b), which had an exceptionally high integration copy number (Fig. 2c). This strain showed an ~7% decrease in growth rate (two-tailed $t$-test $p = 0.001$).

Long-read sequencing on strains containing Constructs 3 and 4 confirmed that the constructs were integrated into the $RPL25$ (YOL127W) locus and that $yEGFP$-$RPL25$ sequences were amplified in tandem repeat structures (Supplementary Figs. 2 and 3–5). The strain expressed the highest level of yEGFP (Construct 4) was sub-cultured in yeast extract-peptone-glucose medium for ~48 generations for stability test (Supplementary Fig. 6). GFP fluorescence levels and population homogeneity did not change, indicating that HapAmp is genetically stable.

**Improving heterologous production of the sesquiterpene trans-nerolidol**. We examined the performance of the HapAmp method using sesquiterpene ($C_{15}$; trans-nerolidol) production. We used a background strain with an upregulated mevalonate pathway for production of terpene precursors (o401R)[35–38]. In this strain, the $GAL80$ repressor gene is disrupted allowing diauxic induction of $GAL$ promoters, which are used to control transgenes.

We constructed a reference strain N401-1 harbouring a multi-copy 2μ plasmid pJT9RFR[39] (Fig. 3a) with overexpression cassettes for farnesyl pyrophosphate synthase ($ERG20$) and nerolidol synthase ($Ac.NES1$). The nerolidol synthase cassette includes a fluorescence-activating and absorption-shifting tag (Y-FAST)[40] and a 2A peptide from Equine rhinitis B virus 1[41] fused to the N-terminus of nerolidol synthase. This allows Y-FAST fluorescence to be used as a proxy for nerolidol synthase expression[39].

The nerolidol synthase expression cassette ($Y$-$FAST$-$2A$-$Ac.NES1$) was cloned into the $RPL25$ insertion vector in the amplification region with three different promoters for replacement of the $RPL25$ promoter; the $ERG20$ expression cassette was cloned at the non-amplification region (Fig. 3b). Colonies with bright Y-FAST fluorescence were selected from the transformation plates. This delivered strains N401-2, N401-3, & N401-4 (promoters $P_{ERG1}$, $P_{PDA1}$, and $P_{BTS1}$, respectively).

Compared to the reference strain N401-1, these three strains exhibited faster growth (Fig. 3c, d), higher Y-FAST fluorescence (Fig. 3f), and higher nerolidol production (Fig. 3h). The $Y$-$FAST$-$2A$-$Ac.NES1$ cassette was successfully amplified in vivo in the three test strains (Fig. 3e).

The reference 2μ plasmid strain harboured 14 copies of the $Y$-$FAST$-$2A$-$AcNES1$ construct, similar to strain N401-3, and higher than that in strain N401-2. However, N401-1 had the lowest $Y$-$FAST$ fluorescence (Fig. 3f). The discrepancy between copy number and fluorescence was due to lack of induction of Y-FAST expression in a large proportion of N401-1 cells (Fig. 3g). In contrast to the 2μ plasmid strain, the strains harbouring the in vivo amplification constructs showed better synchronicity for Y-FAST induction (Fig. 3g N401-3; others not shown). This may contribute to the improved production.

**Improving heterologous production of the monoterpene limonene**. We next tested the system on production of monoterpenes ($C_{10}$). Monoterpene production requires introduction of a dedicated $C_{10}$ geranyl pyrophosphate (GPP) synthase[42]. We have previously used an Erg20p[N127W] mutant[42], which excludes the $C_{15}$ chain from the active site to generate a GPP pool, in combination with targeted degradation of the endogenous $C_{15}$ synthase Erg20p via protein degron tags[35,39] to decrease competition at the $C_{10}$ node by Erg20p and redirect GPP towards monoterpene production. In mevalonate pathway-enhanced strains, this approach delivered less than $100 \text{ mg l}^{-1}$ monoterpene—an order of magnitude below the levels achieved for sesquiterpene engineering.

We used a mevalonate pathway-enhanced strain with the endogenous Erg20p under an auxin-inducible protein degradation mechanism[39] as a background strain to minimise flux competition through the native sterol pathway. Two different promoter constructs were developed for amplification of the limonene synthetic module (Fig. 4a). The amplified region contained a fusion of multiple genes: Y-FAST-2A[39], the maltose-binding protein from $E. coli$ for improved solubility[43], a short linker, limonene synthase from $Citrus limon$[35], a 6*glycerine linker, and the Erg20p[N127W F96W] mutant[42] (which has a higher specific GPP production rate than the Erg20p[N127W] mutant) as a GPP synthase. This fusion construct was under the control of the $GAL2$ promoter from $S. kudriavzevii$[44]. The two constructs were transformed into the $RPL25$ locus in the background strain, delivering strains LIM141M ($P_{PDA1}$) and LIM141MH ($P_{BTS1}$).

For the reference strain, the construct was introduced into the background strain via a 2μ plasmid (Fig. 4a). We characterised four biological replicates (LIM141R representing three biological replicates and LIM141R2 representing one biological replicate; Fig. 4). In this case, 2μ plasmid delivered ~2 copies per genome of the limonene synthase/Y-FAST module (shown by Y-FAST copy number; Fig. 4c). LIM141R, the three biological replicates produced ~$40 \text{ mg l}^{-1}$ limonene (Fig. 4f), the titre same to a previous strain LIM141 expressing limonene synthase and Erg20p[N127W] without gene fusion[39]. However, one biological replicate (LIM141R2, Fig. 4) produced ~$300 \text{ mg l}^{-1}$ limonene. LIM141R2 exhibited faster growth and higher Y-FAST fluorescence levels than other three biological replicates (LIM141R, Fig. 4b, d, e). The improvement in LIM141R2 may be caused by unintended genetic variations.

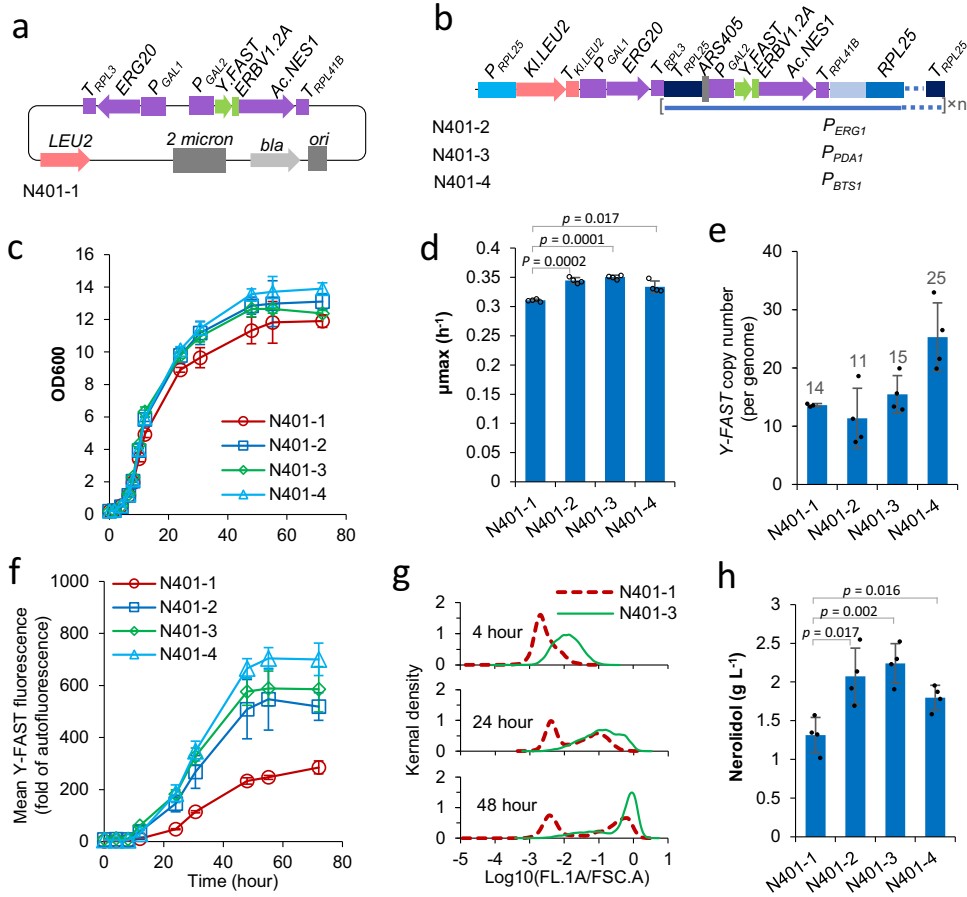

**Fig. 3 Characterisation of nerolidol-producing strains, harbouring nerolidol synthetic genes on a 2μ plasmid (N401-1) or integrated at amplified *RPL25* locus (N401-2, N401-3, and N401-4). a**, **b** Schematic map of genetic vectors used to introduce nerolidol synthetic genes into yeast. **c–h** Strain characterisation in two-phase flask cultivation with 20 g l$^{-1}$ glucose and dodecane overlay. Y-FAST (fluorescence-activating and absorption-shifting tag) fluorescence was measured after 4-hydroxy-3-methylbenzylidene rhodanine (HMBR) with final concentration 20 μM was added to the yeast samples before flow cytometry assay, and is expressed as fold-change of exponential-phase auto-fluorescence of the reference strain GH4[3]. Nerolidol production at 72 h was shown. Kernel density was calculated with bandwidth equal to 0.05. Mean values ± standard deviations are shown (**c–f**, **h**; $N = 4$ independent biological replicates). Two-tailed Welch's *t*-test was used for comparing two groups, and *p* values were shown in **d**, **h**. Source data are provided as a Source Data file.

Harbouring HapAmp limonene synthetic module, both strains LIM141M and LIM141MH produced an order of magnitude more limonene than LIM141R and previous efforts using 2μ plasmids[35,39], with the best production, ~0.95 g l$^{-1}$ limonene at 96 h, by strain LIM141M (Fig. 4f). This titre is 5.6-fold higher than the previous highest titre ever obtained in yeast[45], and ~2-fold higher than the best titres achieved in batch cultivation in *E. coli*[46,47]. Strain LIM141MH showed a slower exponential growth and the lower levels of Y-FAST fluorescence compared to strain LIM141M (Fig. 4b, d, e), despite having more copies of the limonene synthase/Y-FAST module (shown by Y-FAST copy number; Fig. 4c). Both strains also accumulated ~12 mg l$^{-1}$ of the monoterpene alcohol geraniol, which is commonly produced by yeast with an increased GPP pool[35,39]. No farnesol (C$_{15}$ alcohol) or geranylgeraniol (C$_{20}$ alcohol) were accumulated by the strains, indicating that subcellular pools of FPP and the C$_{20}$ geranylgeranyl pyrophosphate (GGPP) were low, and that amplification of limonene synthetic module led to significant redirection of the carbon flux towards monoterpene production.

**Improving heterologous tetraterpenoid lycopene production in yeast**. A three-gene lycopene synthetic module controlled by *GAL* promoters was previously constructed in a 2μ plasmid[37] (Fig. 5a). This construct includes the farnesyl pyrophophase mutant gene

*ERG20$^{F96C}$* which produces GGPP[48], a phytoene synthase[49,50], and a lycopene-forming phytoene desaturase mutant[50]. This plasmid was transformed into a mevalonate pathway-enhanced background strain, generating strain LYC1[37]. This strain accumulated ~5 mg lycopene per gram of biomass in 120-h flask cultivation (Fig. 5b).

The lycopene synthetic module was sub-cloned into both the *PDA1* and *BTS1* promoter *RPL25*-driving HapAmp vectors (Fig. 5a). The resulting constructs were transformed into the same background strain, generating strains LYC4 and LYC5, respectively. Strain LYC4 (*P$_{PDA1}$-RPL25*) accumulated slightly more lycopene than strain LYC1, although the increase was not significant (Fig. 5b). Strain LYC5 accumulated ~25 mg lycopene per gram of biomass, five-fold higher than strain LYC1 (Fig. 5b).

**High-level expression of heterologous proteins in yeast**. *S. cerevisiae* can be used as a platform organism for protein production, including production of pharmaceutical proteins. However, a notorious disadvantage is that heterologous proteins production is not as high as what is achievable with *E. coli* expression systems. The high-level expression in *E. coli* can be attributed to the usage of high-copy-number plasmids (such as the common pET vectors with copy number about ~15–20) and the use of a very strong inducible

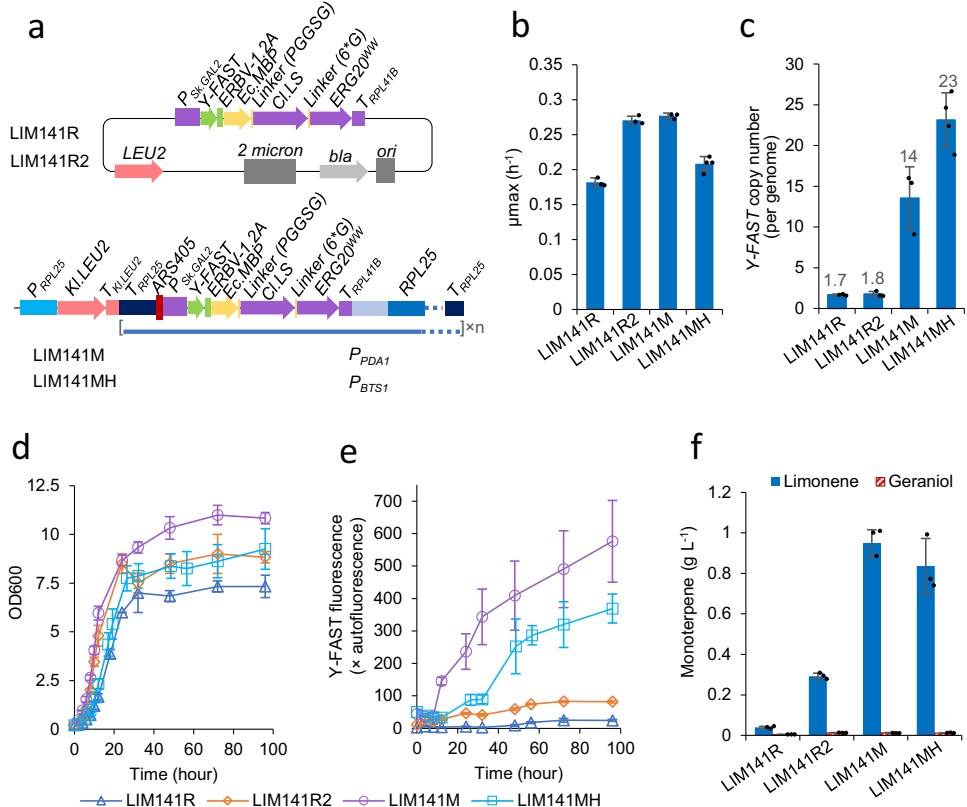

**Fig. 4 Characterisation of limonene-producing strains with limonene synthetic genes on a 2µ plasmid (LIM141R and LIM141R2) or integrated at amplified *RPL25* locus (LIM141M and LIM141MH).** LIM141R2 is one of LIM141R biological replicates. **a** Schematic map of genetic vectors used to introduce limonene synthetic genes into yeast. **b**–**f** Strain characterisation in two-phase flask cultivation with 20 g l$^{-1}$ glucose and dodecane overlay. Synthetic auxin 1-Naphthaleneacetic acid (NAA) was added to 1 mM at the late exponential growth phase (OD > 4). Y-FAST fluorescence was measured after 4-hydroxy-3-methylbenzylidene rhodanine (HMBR) with final concentration 20 µM was added to the yeast samples before flow cytometry assay and is expressed as fold-change of exponential-phase auto-fluorescence of the reference strain GH4[30]. Limonene and geraniol production at 96 h was shown. Mean values ± standard deviations are shown ($N = 3$ independent biological replicates for LIM141R, LIM141M and three independent cultures for LIM141R2 in **b**–**f**. $N = 4$ independent biological replicates in **b**–**e** and three independent biological replicates in **f** for LIM141MH). Source data are provided as a Source Data file.

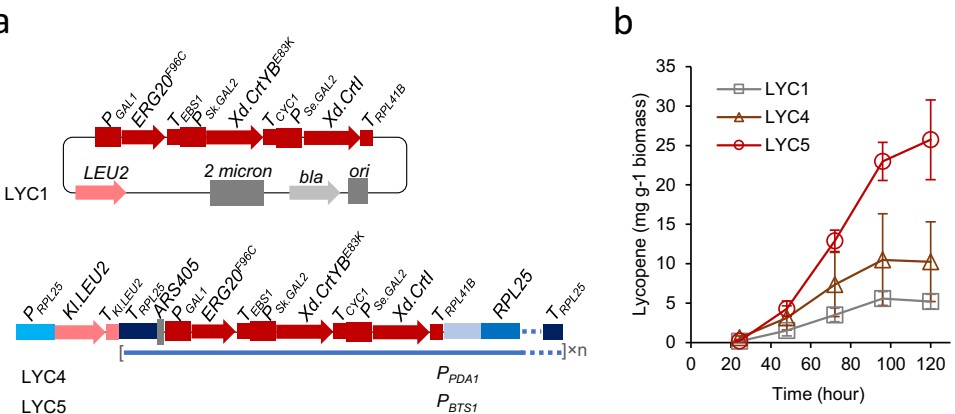

**Fig. 5 Characterisation of lycopene-producing strains with lycopene synthetic genes integrated at amplified *RPL25* locus. a** Schematic maps of genetic vectors used to introduce lycopene synthetic genes into yeast. **b** Lycopene production in flask cultivations. Yeast cells in exponential growth was inoculated into 20 ml MES-buffered YNB medium with 20 g l$^{-1}$ glucose in 125 ml Erlenmeyer flask to start a culture at OD$_{600}$ = 0.2. Mean values ± standard deviations are shown ($N = 4$ independent biological replicates). Source data are provided as a Source Data file.

promoter[51]. We used the $P_{BTS1}$-*RPL25*-driving HapAmp constructs to introduce the *AeBlue* chromoprotein gene[52] (Fig. 6a) or the *EforRed* chromoprotein gene[53]. Blue or pink colonies were obtained on the transformation plates (Supplementary Fig. 7), indicating high-level expression of the chromoproteins.

Having confirmed that the chromoproteins were effective markers, we then inserted a human papillomavirus (HPV) 16 major capsid protein L1 gene after the AeBlue expression cassette (Fig. 6a) to test the system for production of a pharmaceutical protein. For a reference, we cloned AeBlue-and-HPV16-L1

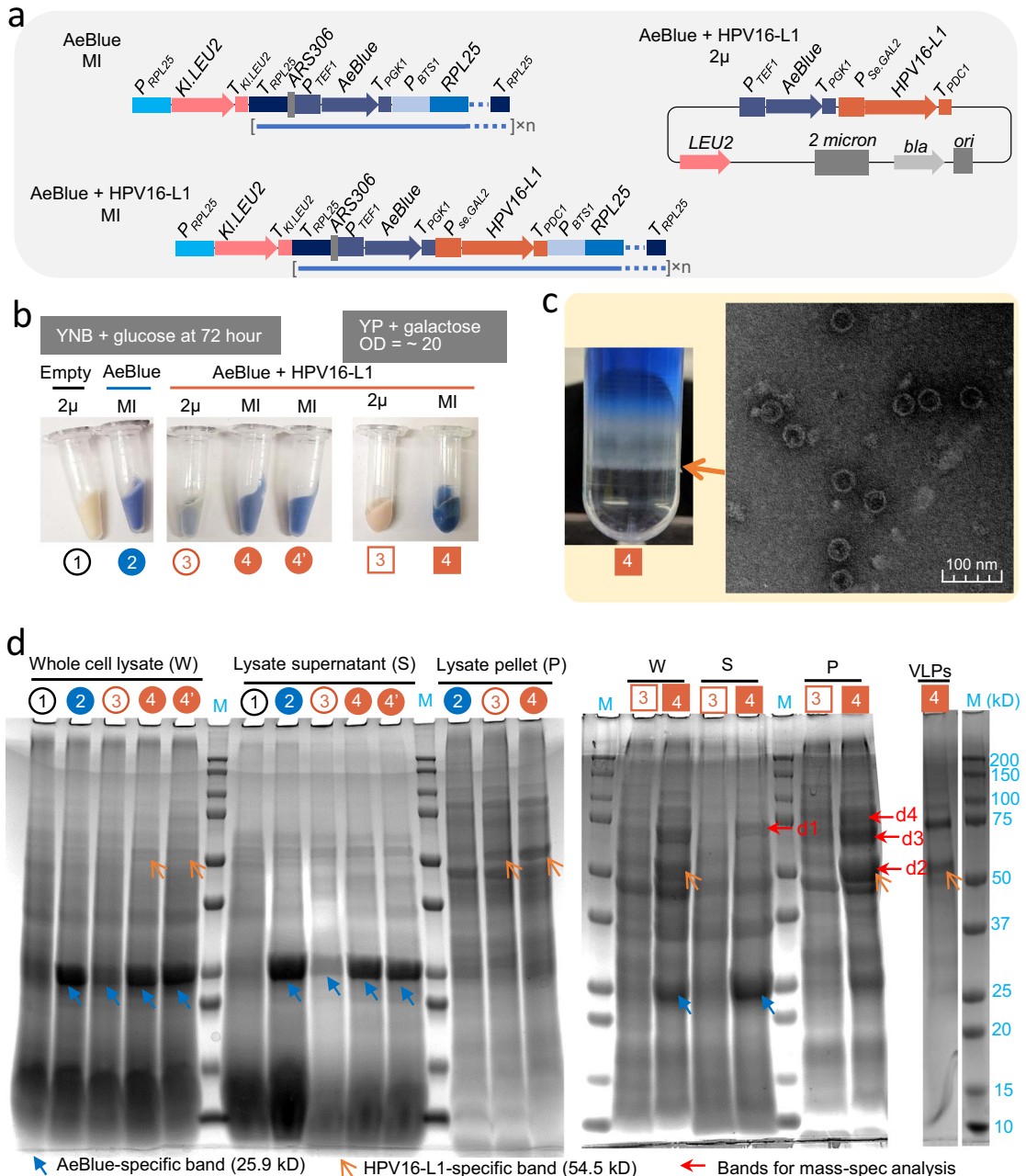

**Fig. 6 Characterisation of the expression of heterologous proteins (AeBlue and HPV16 capsid L1) via multi-copy genome integration (MI) using $P_{BTS1^-}$ RPL25-driven in vivo gene amplification. a** Schematic of genetic vectors used to express AeBlue and HPV16 L1. **b** cells harbouring an empty 2µ, the amplifiable AeBlue construct (MI), AeBlue-and-HPV16-L1 2µ plasmid, and amplifiable AeBlue-and-HPV16-L1 construct (MI). Cells were grown in MES-buffered YNB medium with 20 g l$^{-1}$ glucose and collected at 72 h, or were grown in YP medium with 20 g l$^{-1}$ galactose to $OD_{600} = \sim$20. **c** Ultracentrifugation of the supernatant on an iodixanol gradient to separate a band containing HPV16-L1 virus-like particles (shown by orange arrow), and transmission electron microscopy confirming the presence of HPV16-L1 virus-like particles (VLPs). **d** SDS-PAGE (sodium dodecyl sulphate-polyacrylamide gel electrophoresis) for whole-cell lysates, lysate supernatant, and lysate pellets of yeast samples in **b**, and VLPs sample from **c**. Experimental repetition is not done for **c** and **d**. Numbers in **b–d** are for sample cross-reference. The bands d1, d2, d3, and d4 are analysed using a LC-MS/MS-based proteomic method (Supplementary Method 1), and the data are available in Supplementary Data 1 (d1), Supplementary Data 2 (d2), Supplementary Data 3 (d3), and Supplementary Data 4 (d4). Source data for VLPs in **d** are provided as a Source Data File.

expression cassettes into a yeast 2µ plasmid (Fig. 6a). To compare the efficiency of protein production in different systems, an empty 2µ plasmid, the AeBlue-and-HPV16-L1 2µ plasmid, the *RPL25*-amplifiable AeBlue construct, and the *RPL25*-amplifiable AeBlue-and-HPV16-L1 construct were transformed individually into CEN.PK (*gal80Δ*). The four resulting strains were grown in MES-buffered YNB medium with 20 g l$^{-1}$ glucose aerobically for

72 h. Cells with multi-copy integration of the AeBlue expression cassette showed a strong Tibetan blue colour, while cells with an empty cassette were milky white colour (Fig. 6b). The cells with 2µ plasmid containing AeBlue + HPV-L1 expression cassettes were a faint blue colour, whereas the cells with multi-copy integration of AeBlue + HPV-L1 expression cassettes displayed the strong Tibetan blue colour (Fig. 6b). This indicated superior

expression capacity from the in vivo amplification method for multi-copy genome integration, compared to conventional 2μ plasmid method.

SDS-PAGE of whole-cell and soluble protein extracts showed bands at ~25 kD (AeBlue molecular weight) in all samples, with much stronger bands observed in the multi-copy integration strain samples than in the 2μ plasmid strain samples (Fig. 6d). In the multi-copy integration strains, these bands represented ~3% of whole-cell protein, suggesting heterologous protein expression in yeast may reach the levels often obtained in E. coli.

A second strong band at ~50 kD band (HPV16-L1 molecular weight) was observed in samples from cells expressing HPV-L1, although it was not as distinct at the putative AeBlue band (Fig. 6d). This may be due to the use of the Se.GAL2 promoter, which is not fully induced in the ethanol phase, in these constructs compared to the constitutive ALD6 promoter used for the AeBlue expression cassette. Again, the bands in the multi-copy integration strain samples were stronger than the 2μ plasmid samples. Surprisingly, considering that HPV16-L1 is a soluble protein[54], these bands were not distinguishable in lysate supernatant samples.

To fully induce the Se.GAL2 promoter for HPV16-L1 expression, we attempted to grow the plasmid and integration strains harbouring HPV16-L1 in synthetic minimal medium (YNB) with ethanol or galactose as the carbon source. However, these cultivation conditions were lethal for the multi-copy-integration cells. We then grew the cells in rich (yeast-peptone (YP)) medium with $20 \, g \, l^{-1}$ galactose as the carbon source. Under these conditions, AeBlue expression from 2μ plasmid was not observable by visual examination (Fig. 6b) or SDS-PAGE (Fig. 6d). This may be due to loss of 2μ plasmid in the rich medium. In contrast, strong AeBlue-specific and HPV16-L1-specific bands were seen in whole-cell lysate and lysate supernatant samples from the cells with multi-copy integration constructs. This further confirmed that HPV16 L1 capsid protein is insoluble in yeast in our system. Attempts to solubilise HPV16-L1 L1 capsid protein were unsuccessful (data not shown). Despite being unable to detect HPV16-L1-specific bands in lysate supernatant (Fig. 6d), we could still separate properly assembled virus-like particles (VLPs) by ultracentrifugation of lysate supernatant (Fig. 6c). SDS-PAGE examination of VLP components purified from ultracentrifugation showed a HPV16-L1-specific band at ~50 kD (Fig. 6d; Lane VLPs:4). TEM images of the VLPs showed their diameter was around 40 nm (Fig. 6c), consistent with previous literature[55].

In SDS-PAGE results, we observed strong bands in the lysate supernatant sample (band d1) and lysate pellet samples (bands d2, d3, and d4) (Fig. 6d). LC-MS/MS-based proteomics was used to analyse the protein composition in these four bands (Supplementary Data 1–4). The top hit protein in the ~50 kD band (band d2) was the HPV16 L1 capsid. Interestingly, the top hit proteins in other three bands (d1, d3 and d4) were yeast chaperones. In bands d1 and d3, the top hit proteins were HSP70 family chaperone Ssa1, and in bands d4, the top hit protein was HSP90 family chaperone Hsc82. We therefore hypothesised that insoluble expression of HPV16-L1 caused upregulation of yeast chaperones, and HPV16-L1, HSP70 chaperones, and HSP90 chaperones might exist in insoluble forms. However, it would require further systematic examination to get a better understanding of these phenomena.

In summary, although some insoluble expression of the HPV16 L1 was observed, our results both with chromoprotein AeBlue and the HPV16 L1 showed that multi-copy gene integration via HapAmp method can lead to heterologous protein overexpression in yeast to the high levels that are commonly seen in E. coli expression systems.

## Discussion

Here, we developed a genetic engineering method to integrate multiple copies of heterologous gene(s) into the yeast genome using in vivo gene amplification driven by a haploinsufficient gene (HapAmp). The functional strength per copy of a haploinsufficient gene is strongly associated with growth fitness, which can be exploited as an evolutionary force to drive gene amplification. Decreased expression level provides an evolutionary force that drives amplification of linked haploinsufficient and heterologous genes, so that cells are growth competitive. We exemplified the application of this method to improve production of different types of terpene products. We also showed that our method enabled high-level expression of heterologous protein in yeast, at levels similar to that achieved in E. coli for protein production.

This method presents three main advantages for the introduction of heterologous genes via genome integration. Firstly, integration copy number can be titrated by altering expression dosage per copy of haploinsufficient gene. Expression level can be reduced by a variety of methods. Here, we tested two approaches: (1) replacing the gene promoter with a weaker promoter (Figs. 2–4), and (2) using non-preferred codons (Fig. 2). In these experiments, we observed a range of between 4 and 47 copies, with an inverse relationship between promoter strength and copy number. We characterised a range of weak promoters here (Supplementary Fig. 8) and in previous work[3] that can be applied to decrease gene dosage. In addition to promoter strength and codon usage, other approaches could be used to decrease expression dosage, including engineering the kozak sequence and/or the 5′-mRNA structure. These genetic tools add engineering flexibility to modify copy number for this HapAmp method in yeast.

Secondly, the maintenance of integration is auto-selectable: selection pressure is provided from the dosage sensitivity of the haploinsufficient gene, which is linked to the gene of interest and is maintained to support normal growth rates. This means that no antibiotics or modification of other environmental conditions in the culture are required to provide ongoing selection pressure for maintenance of the gene of interest. Compared to use of a 2μ plasmid, this method provides more stable expression of heterologous proteins in yeast (Fig. 6b). In addition, it does not require chemical induction for amplification[2,15].

Thirdly, the presence of multiple haploinsufficient genes means that many different loci are available for engineering gene amplification. We demonstrated the method using RPL25 and SEC23 as the driving gene. We further characterised the promoter strength of fifteen additional haploinsufficient genes (Supplementary Fig. 8) that can also be used to drive gene amplification.

Initial integration of the genes of interest uses standard yeast transformation procedures by selection of an auxotrophic or antibiotic marker (e.g. LEU2 or hphMax in Figs. 2–6). Upon transformation, we observed a variable proportion of false clones (not expressing the gene of interest) on the transformation plates (Supplementary Figs. 1 and 5). We presume that, in these cases, either spontaneous mutations have provided the yeast an alternative mechanism to recover growth rate, or the gene of interest was not correctly integrated into the target locus. Use of visual markers (fluorescent proteins or chromoproteins) can facilitate the selection of correct clones with amplified constructs. In the absence of such visual markers, characterisation or verification of a pool of clones would be necessary to select clones with multi-copy integration of heterologous genes. Further optimising the genetic background of the yeast strains used, such as eliminating the non-homologous end joining mechanism to decrease non-homologous gene integration, might be useful to eliminate rate of false positives for the current method.

The HapAmp method successfully improved production of heterologous terpenes including the $C_{15}$ sesquiterpene nerolidol (Fig. 3), the $C_{10}$ monoterpene limonene (Fig. 4), and the $C_{40}$ tetraterpene lycopene (Fig. 5). Production of $C_{15}$ terpenes in yeast is typically relatively straightforward, with gram per litre titres achievable[39,56]. This is likely because the $C_{15}$ precursor, FPP, is produced in yeast naturally to deliver sterol pathway products required for yeast growth. In addition, sesquiterpene synthases have reasonably good catalytic properties, making them more competitive to access FPP. Production of $C_{10}$ monoterpenes, however, has historically been very challenging. This is due to both a dearth of $C_{10}$ precursors[57] and the poor catalytic properties of many monoterpene synthases[45,58]. These limitations have previously restricted published titres of monoterpenes to mg $l^{-1}$ in flask cultivation[35,39,45,59]. Here, we have achieved g $l^{-1}$ titres (Fig. 4) in a single engineering step using a high mevalonate pathway flux strain with an introduced GPPS and targeted degradation of FPPS to decrease competition at the $C_{10}$ pathway node. We believe this is the highest titre achieved in metabolically engineered microbes in a flask cultivation with 20 g $l^{-1}$ glucose as carbon source reported to date.

Interestingly, one replicate of the monoterpene control strain produced ~300 mg $l^{-1}$ limonene, in comparison to the other three replicates which produced ~40 mg $l^{-1}$ limonene, despite the plasmid copy number being the same in all four replicates (Fig. 4). This suggests that an unintended mutation has arisen in this strain which affects limonene production positively. The source of this variation will be examined in future work and may form the basis of further engineering efforts.

We observed a tight correlation between gene copy number and GFP fluorescence (Fig. 2); however, this relationship breaks down for the different terpene products, resulting in variable improvement ratios. This is most likely due to the fact that the relationship between the GFP peptide and its fluorescence is very close and does not rely on other factors such as substrate and cofactor availability—whereas the terpene synthases are enzymes and subject to more influences on their behaviour. In addition, variable metabolic burden caused by overexpression of terpene synthases or other physiological perturbations in metabolically engineered systems may affect the relationship between copy number and product titre. For products, limonene production improvement was ~24-fold, whereas nerolidol improvement was 1.7-fold, and lycopene improvement was 5-fold. However, we always obtained a higher titre by in vivo gene amplification. In particular, for monoterpenes, insufficient catalytic efficiency of terpene synthase is a significant bottleneck for production of heterologous terpenoids in yeast. Increasing copy number via insertion of tandem repeats at the same locus combined with screening for improved production[56] or introduction of additional expression cassettes at separate loci[43] has been used to overcome this bottleneck previously. However, these approaches require complex cloning and extended experimental timelines to deliver the desired improvements. The HapAmp system provides a faster and simpler method to achieve superior results.

We tested several constructs ranging up to three expression cassettes (lycopene pathway: insert size of 7917 bp). We have not sought to test the maximum cargo size for this approach. However, we did not observe a clear relationship between size of the insert ('cargo') and copy number amplification, suggesting that even larger inserts may be possible for the technique.

In addition to its application in metabolic engineering, we also examined the potential of HapAmp for increasing heterologous protein production. Using chromoprotein AeBlue and the HPV16 L1 capsid protein as examples (Fig. 6), we demonstrated that in S. cerevisiae, heterologous protein could be produced at levels commonly seen in E. coli. AeBlue was expressed in soluble form, whereas HPV16 L1 capsid protein was primarily expressed in insoluble form. Insoluble expression of HPV16 L1 capsid protein has been reported in E. coli[60–62] but not in S. cerevisiae. In E. coli, N-terminal truncation[60,61], use of a fusion partner[61], and overexpression GroEL/GroES chaperones[62] (which accept broader substrates than cytosolic chaperones in S. cerevisiae[63,64]), improved soluble expression of HPV L1 capsid proteins. These strategies might also improve soluble expression of HPV capsid proteins in yeast.

The HapAmp method should be applicable in other industrially relevant chassis organisms that have haploinsufficient genes. A potential haploinsufficient gene may encode essential components of the machineries for protein synthesis and transportation or other essential cell structures[28]. Putative haploinsufficient genes can be identified by comparative genomics and confirmed by testing growth fitness in association with expression dosage of a gene. For diploid organisms, this can be done by disrupting one allele and integrating the amplifiable construct at the other allele locus, or by simultaneously integrating the amplifiable constructs at both alleles. In addition, native non-homologous end joining mechanisms can be diminished/disrupted to improve the successful rate of amplification of genes of interests[65]. A nuclease-mediated DNA double-chain break like CRISPR[66] could also be used to assist the integration of the amplifiable construct. This may avoid the use of a selectable marker in the gene amplification construct.

## Methods

**Plasmid and strain construction**. Plasmids used in this work are listed in Supplementary Data 5, and strains are listed in Supplementary Data 6. Primers used in polymerase chain reaction (PCR) and PCR performed in this work are listed in Supplementary Data 7. Plasmid construction processes are listed in Supplementary Data 8. Yeast strain construction processes are listed in Supplementary Data 9. A LiAc/SS carrier DNA/PEG method[67] was used for yeast transformation.

**Yeast cultivation**. For characterisation of yEGFP-expressing strains, yeast cells from glycerol stocks were streaked on YNB-glucose agar, which comprised of 6.9 g $l^{-1}$ yeast nitrogen base without amino acids (YNB, FORMEDIUM#CYN0402) with pH adjusted to 6.0 using sodium hydroxide solution, 20 g $l^{-1}$ glucose, and 20 g $l^{-1}$ agar. MES-buffered YNB-glucose medium was used in following cultivation, which comprised of 19.5 g $l^{-1}$ 2-(N-morpholino)ethanesulfonic acid (MES), 6.9 g $l^{-1}$ YNB, 20 g $l^{-1}$ glucose, and its pH was adjusted to 6.0 with ammonia hydroxide solution. For the growth in flask, seed cultures grown to the exponential phase ($OD_{600} \leq 4$) were inoculated into 20 ml MES-buffered YNB-glucose medium in 125 ml Erlenmeyer flasks to start the cultivation in a 200 rpm 30 °C incubator. For the growth in 96-well microplate, yeast cells were grown in YNB-glucose medium (6.9 g $l^{-1}$ YNB, 20 g $l^{-1}$ glucose, pH 6.0) for about 20 h to stationary phase in a 350 rpm 30 °C incubator to prepare seed culture. Seed culture (5 μl) was inoculated into 100 μl MES-buffered YNB-glucose medium to prepare Culture 1. Culture 1 (2 μl) was inoculated into 100 μl MES-buffered YNB-glucose medium to prepare Culture 2. Culture 2 was incubated in a 350 rpm 30 °C incubator overnight for analysis of yEGFP fluorescent in the cells grown to the exponential growth phase, and Culture 1 for two nights for analysis in the cells grown to the ethanol growth phase.

For characterisation of nerolidol/limonene-producing strains, dodecane-overlayed two-phase flask cultivation was used. Yeast cells from glycerol stocks were streaked on YNB-high-glucose agar, which contained 6.9 g $l^{-1}$ YNB (pH 6.0), 200 g $l^{-1}$ glucose, and 20 g $l^{-1}$ agar. Before initiating the two-phase flask cultivation, cells were pre-cultured in MES-buffered YNB-20 g $l^{-1}$ glucose to exponential phase ($OD_{600}$ between 1 to 4) and collected by centrifugation. Collected cells were then resuspended in fresh fermentation medium. To initiate the cultivation, appropriate volumes of pre-cultured cells were transferred to MES-buffered YNB medium with 20 g $l^{-1}$ glucose to an initial $OD_{600}$ of 0.2 in a total volume of 23 ml medium in a 250 ml flask, and 2 ml sterile dodecane was added after inoculation. In the first 12 h of cultivation, 3 ml culture was sampled for growth curve measurement. Dodecane was sampled and stored at −80 °C for terpene analysis.

Flask cultivations for lycopene-producing strains were prepared as the flask cultivation used for yEGFP-expressing strains. For chromoprotein/HPV16 L1-expressing strains, yeast cells grown overnight in 5 ml MES-buffered YNB-glucose medium were inoculated into 20 ml fresh MES-buffered YNB-glucose medium or 20 ml YP-galactose (20 g $l^{-1}$ peptone, 10 g $l^{-1}$ yeast extract, and 20 g $l^{-1}$ galactose) to start characterisation cultures.

**Flow cytometry**. A BD Accuri™ C6 flow cytometer (BD Biosciences, USA) was used for fluorescence analysis in single cells. Cells expressing yEGFP were sampled and directly used for characterisation of the yEGFP fluorescence. Cells expressing Y-FAST was sampled and mixed with 20 μM HMBR (synthesised and prepared in 2 mM stock in dimethyl sulfoxide[40]) before analysis. Debris particles were excluded through an FSC.H threshold with the threshold value of 250,000. A 488 nm laser was used to excite GFP and Y-FAST fluorescence. The detector equipped with a 530/20 bandpass filter was used to monitor the fluorescence (FL1.A). For each sample, 10,000 events were recorded. A BD Csampler software (BD Accuri C6 software version 1.0.264.21) were used to extract mean values of FSC.A, SSC.A, and FL1.A. The fluorescence level of GFP and Y-FAST was expressed as the fold of a background fluorescence in the exponential grown phase cells of strain GH4[3].

**Metabolite analysis**. HPLC analysis was performed by the Metabolomics Australia (Queensland node) using a previously described method[68]. In brief, an Agilent 1200 HPLC system and a Thermo Fisher Chromeleon Chromatography Data System software were used. Dodecane samples in some cases were diluted with dodecane before HPLC analysis. For HPLC analysis, 5 μl dodecane samples (or standards prepared in dodecane) were mixed with 200 μl ethanol, and 20 μl mixture was injected and separated with a guard column (SecurityGuard Gemini C18, Phenomenex PN: AJO-7597) and a Zorbax Extend C18 column (4.6 × 150 mm, 3.5 μm, Agilent PN: 763953-902). The mixture of solvent A (water) and solvent B (45% acetonitrile, 45% methanol, and 10% water) was used to elute the analytes with a linear gradient (from 0–24 min, 5–100% solvent B; from 24–30 min, 100% solvent B; from 30.1–35 min, 5% solvent B).

For lycopene measurement, yeast cells were collected and resuspended in 200 μl 2 M l$^{-1}$ sodium hydroxide and vortexed with 200 mg glass bead and 1 ml hexane for at least 10 min. Lycopene molar extinction coefficient ($182 \times 10^3$) at 471 nm was used to calculate lycopene concentration[69]. In some cases, lycopene extracts were diluted with hexane to make the absorbance reading <0.6.

**Protein purification**. Yeast cells were homogenised by vortexing with glass beads for 15 min in phosphate-buffered saline (PBS) buffer plus 2 mM ethylenediaminetetraacetic acid. Whole-cell lysates, lysate supernatants, and lysate pellets were examined by sodium dodecyl sulphate-polyacrylamide gel electrophoresis analysis on Mini-PROTEAN® Precast Gels (Bio-rad).

The lysis was followed by centrifugation at $18,000 \times g$ for 30 min to pellet the cellular debris. The soluble fraction was then loaded on top of a gradient made of 1 ml of 20% Iodixanol/PBS buffer, 1 ml of 30% Iodixanol/PBS and 1 ml of 40% Iodixanol/PBS in a Thinwall Ultra-Clear Tube (Beckman Coulter, Indianapolis, USA) and subjected to ultracentrifugation for 2 h 30 min at $150,000 \times g$ on a SW41 Ti rotor or a using a Beckman Optima L-100XP ultracentrifuge (Beckman Coulter, Indianapolis, USA). A band containing the VLPs encapsulating protein was extracted using a 1 ml syringe by poking a whole through the tube. Bradford was used to measure protein concentration and sample was further examined on TEM and purity confirmed on Mini-PROTEAN® Precast Gels (Bio-rad).

**Transmission electron microscopy**. Samples containing purified VLPs of 0.1 mg ml$^{-1}$ were applied to formvar/carbon coated grids (ProSciTech Pty Ltd, Australia) and incubated for 2 min. Grids were then washed with 40 μl of distilled water for 30 s twice, and then stained with 20 g l$^{-1}$ uranyl acetate for 1 min, after being blotted on filter paper. Images were taken on a HITACHI HT7700 transmission electron microscope at accelerating voltage of 80 keV at the Centre for Microscopy and Microanalysis.

**Genome sequencing**. Yeast genomic DNA was extracted using MagAttract HMW DNA Kit (Qiangen) with a modified protocol. Yeast cells (20 ml, OD$_{600}$ around 10) were washed once using PBS buffer and resuspend in 2 ml 1 M sorbitol solution. Yeast cell walls were digested by adding 30 U Zymolyase-20T (nacalai, Japan; 1 U per μl in 1* PBS containing 100 mM DTT and 50% v/v glycerol) at 30 °C for 30 min. Yeast protoplast cells were collected and resuspended in 300 μl Buffer AL (MagAttract HMW DNA Kit) by pipetting using wide bore pipette tips, and then 360 buffer ATL (MagAttract HMW DNA Kit) was added and mixed. Following this, protocol provided in MagAttract HMW DNA Kit (Qiangen) was adopted including digestion by Proteinase K and Rnase A and purification using magnetic beads. Genomic DNA was eluted using 400 μl Buffer AE (MagAttract HMW DNA Kit) and treated using 100 μl tris-saturated phenol (pH 8.0, Ameresco) by flickering and 100 μl chloroform was added and mixed. Upper-layer water phase was collected after centrifuging at $17,000 \times g$ for 5 min and mixed with 1 ml ethanol. Magnetic beads (MagAttract HMW DNA Kit) were used to purify genomic DNA with twice 70% ethanol wash and elution in 50 μl water. Concentration of genomic DNA was quantified using Qubit Fluorometer and Qubit dsDNA BR Assay Kit (Thermo Fisher). Genomic DNA (500 ng) was used to prepare genome sequencing library using Rapid Barcoding Kit (SQK-RBK004, Oxford Nanopore) and sequenced using R9 flowcell MIN106D and MinION Mk1C (Oxford Nanopore). High-accurate base-calling was performed using ont-guppy-for-mk1c (version 4.2.3) installed MinION Mk1C (MinKNOW version 20.10.6). Galaxy Australia online server was used for data processing[70]. Collapse Collection (Galaxy Version 5.1.0) was used to combine fastq dataset into a single file. Nanoplot was used for

statistical analysis of MinION reads[71]. Canu assembler was used for genome sequence assembly[72]. Maker (Galaxy Version 2.31.11) was used to collect annotation evidence with input of *S. cerevisiae* gene sequences and heterologous gene sequences as ESTs input file[73]. miniMap2 was used to align trimmed reads outputted by Canu assembler against contigs outputted by Canu assembler[74]. JBrowse (version 1.16.10-desktop)[75] and Integrative Genomics Viewer (version 2.8.13)[76] were used to illustrate genome structure and read alignment.

**Reporting summary**. Further information on research design is available in the Nature Research Reporting Summary linked to this article.

## Data availability

MinION whole genome sequencing raw-read data are achieved in NCBI BioProject database with submission ID PRJNA688119. Processed data for MinION genome sequencing are achieved in Zenodo (https://zenodo.org/record/6378077#.YnPhi9rMI2w; https://doi.org/10.5281/zenodo.6378077). Plasmids used in this study are available on request or on Addgene (Addgene IDs: 185870-185894) (https://www.addgene.org/Claudia_Vickers/). Source data are provided with this paper.

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

## Acknowledgements

B.P. and this research were supported by a CSIRO synthetic biology future science fellowship and the University of Queensland. B.P. and C.E.V. acknowledge current support from Australian Research Council Centre of Excellence in Synthetic Biology and Queensland University of Technology. Metabolite analysis was performed by Dr Manual Plan in Metabolomics Australia (Bioplatform Australia) Queensland Node. HPLC-MS/MS analysis was performed by Dr Lian Liu in Bioplatform Australia Queensland Node. Yeast strains in this study derives from CEN.PK background strains, which were provided by EUROSCARF (Scientific Research and Development GmbH, Germany) under a non-commercial licence. The authors also acknowledge the facilities, and the scientific and technical assistance, of the Australian Microscopy and Microanalysis Research Facility at the Centre for Microscopy and Microanalysis, The University of Queensland.

## Author contributions

B.P. and C.E.V. contributed to the conception of the project. B.P. designed and performed experiments. L.E. participated in protein purification and analysis. Z.L., Q.S., and L.C.C. participated in strain construction and characterisation. B.P. drafted manuscript. C.E.V. revised manuscripts. G.D., M.T., C.S., and C.B.H. provided advice opinions and participated in manuscript revision. C.E.V., G.D., M.T., and C.S. participated in the support and coordination of the project. All authors contributed to result analysis and discussion.

## Competing interests

The University of Queensland has filed two Australian provisional patents on the methods for gene amplification to claim the intellectual property (Inventors: B.P. and C.E.V. Australian Patent Application numbers: 2022900699 and 2022901094). C.E.V. has a financial interest in Provectus Algae. Other authors declare no competing interests.
