## [Peer Review File · Nature Communications]

An in vivo gene amplification system for high level expression in *Saccharomyces cerevisiae*Reviewers' Comments:

Reviewer #1:

Remarks to the Author:

The manuscript by Peng et al. proposes a method for in vivo gene amplification for high level expression in yeast. This is a novel strategy for reconstruction of biosynthetic pathways and high-level expression of heterologous proteins in yeast host, which addresses the current limitation and will clearly simplify the metabolic engineering efforts. The manuscript is well written and designed. In the first part of this study, the authors present the HapAmp system. The authors take advantage of the expression level of haploinsufficient genes in yeast as an evolutionary pressure associated with growth fitness to drive gene amplification. Employing weaker promoters or non-preferred codons to reduce expression level of two known haploinsufficient genes, they achieve integration of up to 47 copies of target DNA fragments into the yeast genome. Applications of HapAmp are demonstrated for production of different classes of terpenoids of varied degree of difficulty including nerolidol, limonene and lycopene. Notably, the authors obtain ~1 g/L of limonene in shake flasks by HapAmp, which usually require extensive engineering steps and scale-up processing. Moreover, the HapAmp applicability to protein production is exemplified. Overall, I consider the work is highly interesting, timely and of broad interest for the biomanufacturing sector.

However, I have to raise some comments:

The comparison of limonene production from a 2 μ plasmid versus the HapAmp system is missing and the 20-fold improvement obtained here are based on previous yields obtained by the group. While I am well aware of the importance of the high limonene yield achieved through such a quite straightforward method, the results must be clearly demonstrated.

I am wondering why the authors chose HPV16 L1 expression, particularly knowing about the insolubility issues reported in bacteria.

Line 34: the detail of cultivation "on 20 g L⁻¹ glucose" is not necessary as this is standard growing procedure for yeast.

Line 48: Please mention the drawbacks of available methods to clarify the need of this study.

Line 51: rDNA is not explained

Line 119: The tested yeast promoters are stated as "data not shown", however, some of them are presented in supplementary fig. 5 and some were characterized in previous work. So, I consider a list of the characterized promoters would be useful in the supplementary information.

Line 120-126: A supplementary table to describe constructs 1-8 would be useful here, as it is difficult to go back and forth to follow the construct description.

Line 171: R is missing in "PBTS1-PL25". Moreover, the name of the combination promoter/hap gene versus construct number is not consistently used in this chapter and confusing. Both names should be provided for clarity.

Line 178: Please specify the genotype of the background strain with an upregulated mevalonate pathway used as it is difficult to identify it in the 3 references.

Line 412: Lycopene is not a C30 triterpene, but a tetraterpene (C40) as correctly stated in the abstract.

The numbering code in figure 6 is not explained.

The methods for LC-MS/MS-based proteomic analysis is missing.

Reviewer #2:

Remarks to the Author:

The authors present a potentially interesting approach for multi-copy integrations and pathway amplifications. There are a few aspects that are lacking in the current manuscript to gain a better appreciation for this proposed approach.

A very common approach in the field for multi-copy integrations in yeast is via that pITY system developed by Parekh et al from the Wittrup group. This system integrates into the Ty element repeats

in yeast and can likewise achieve very high number of stable copies. How does this system reported here compare? A head-to-head experimental comparison of one of the targets (perhaps one of the terpenes) should be included here otherwise the report does not demonstrate novelty.

Stability is an important factor on multi-copy integrations. The authors must delve into this effect with multiple generations and subcultures to watch for changes in the copy number as a function of long-term culturing.

The authors show a very nice correlation between copy number and expression when it comes to GFP, but this correlation breaks down when looking at most of the products. A further discussion here is warranted.

What is the limit of "cargo" size that may be put into this system? How large of a pathway/segment of DNA can be used? More complex pathways should be investigated.

Reviewer #3:

Remarks to the Author:

The manuscript of Peng et al aims to address an important issue in eukaryotic microbial cell factories, that of high level, stable heterologous gene expression. It appears that the capacity of cells to produce enzymes has not been exhausted in yeast. To achieve stability of gene expression and economic feasibility of the production platform, the expression cassettes usually propagated and expressed in plasmids require transfer to the genome in multiple copies. Current approaches to expanding gene copy numbers have been somewhat unsatisfactory.

The proposed new approach takes advantage of haploinsufficiency of certain endogenous genes, to force gene amplification. This is accomplished by substituting the endogenous promoter of the gene with much weaker promoters, which cause the haploinsufficient phenotype. Linking this gene with the expression cassette (flanked by the duplicated terminator sequence and an ARS) causes the simultaneous amplification of the cassette to compensate for the growth defects caused by low gene expression levels. This approach was tested with two haploinsufficient genes and eight different weak promoter constructs. Amplified copies varied between 5-47 copies and GFP reporter levels reaching in the highest case 92-fold over single copy integrants. The system was validated with three different terpene synthases and a viral protein.

Overall, it is a well written manuscript that has the potential to move the field of Microbial Cell Factories closer to expanded industrial utilization.

Questions:

1. In different constructs, the ARS used are different. Is the nature of the ARS expected to play a role in cassette amplification?
2. Could the authors test levels of expression of the two haploinsufficient genes subsequent to amplification and compare them with pre-amplification levels, as well as the wild type ones?
3. Figure 3b It would be helpful to the readers to annotate the region of amplification on the figure, and include the far-right T sequence. The same for figure 4a and 5a.
4. L395 post-diastolic-shift. Is this a mistake?

RESPONSES to REVIEWER COMMENTS

Reviewer #1 (Remarks to the Author):

The manuscript by Peng et al. proposes a method for in vivo gene amplification for high level expression in yeast. This is a novel strategy for reconstruction of biosynthetic pathways and high-level expression of heterologous proteins in yeast host, which addresses the current limitation and will clearly simplify the metabolic engineering efforts. The manuscript is well written and designed. In the first part of this study, the authors present the HapAmp system. The authors take advantage of the expression level of haploinsufficient genes in yeast as an evolutionary pressure associated with growth fitness to drive gene amplification. Employing weaker promoters or non-preferred codons to reduce expression level of two known haploinsufficient genes, they achieve integration of up to 47 copies of target DNA fragments into the yeast genome. Applications of HapAmp are demonstrated for production of different classes of terpenoids of varied degree of difficulty including nerolidol, limonene and lycopene. Notably, the authors obtain ~1 g/L of limonene in shake flasks by HapAmp, which usually require extensive engineering steps and scale-up processing. Moreover, the HapAmp applicability to protein production is exemplified. Overall, I consider the work is highly interesting, timely and of broad interest for the biomanufacturing sector.

Response: We thank the Reviewer for recognising the significance of this work and appreciate the comments for improvement.

However, I have to raise some comments:

Q1. The comparison of limonene production from a 2 μ plasmid versus the HapAmp system is missing and the 20-fold improvement obtained here are based on previous yields obtained by the group. While I am well aware of the importance of the high limonene yield achieved through such a quite straightforward method, the results must be clearly demonstrated.

Response: We have added the 2 μ plasmid reference strain. We characterised four biological replicates. Three produced ~40 mg L⁻¹, and one produced ~300 mg L⁻¹ limonene. For all four replicates, the 2 μ plasmid was maintained at ~2 copies of limonene synthase module (limonene synthase + YFAST) per genome copy as measured by real-time PCR. From these results, we believe that outlier (~300 mg L⁻¹ limonene) most likely has a mutation that increases limonene production. We will examine this further in future work to try and identify the cause.

Despite the behaviour of the outlier, the key finding is that the reference experiment supports the conclusion that the HapAmp system can contribute to significant improvement in metabolic engineering applications.

We have added the new data into Figure 4 and made the following revision in the main text:

For the reference strain, the construct was introduced into the background strain via a 2 μ plasmid (**Figure 4a**). We characterised four biological replicates (LIM141R representing three biological replicates and LIM141R2 representing one biological replicate; Figure 4). In this case, 2 μ plasmid delivered ~2 copies per genome of the limonene synthase/Y-FAST module (shown by Y-FAST copy number; **Figure 4c**). LIM141R, the three biological replicates produced ~40 mg L⁻¹ limonene (**Figure 4f**), the titre same to a previous strain LIM141 expressing limonene synthase and Erg20p^{N127W} without gene fusion (36). However, one biological replicate (**LIM141R2, Figure 4**) produced ~300 mg L⁻¹ limonene. LIM141R2 exhibited faster growth and higher YFAST fluorescence levels than other three biological

replicates (LIM141R, **Figure 4b, 4d, and 4e**). The improvement in LIM141R2 may be caused by unintended genetic variations.

Harbouring HapAmp limonene synthetic module, both strains LIM141M and LIM141MH produced an order of magnitude more limonene than LIM141R and previous efforts using 2 μ plasmids (33,36), with the best production, ~0.95 g L⁻¹ limonene at 96 hr, by strain LIM141M (**Figure 4f**). This titre is 5.6-fold higher than the previous highest titre ever obtained in yeast (42), and ~2-fold higher than the best titres achieved in batch cultivation in *E. coli* (43,44). Strain LIM141MH showed a slower exponential growth and the lower levels of Y-FAST fluorescence compared to strain LIM141M (**Figure 4b, 4d & 4e**), despite having more copies of the limonene synthase/Y-FAST module (shown by Y-FAST copy number; **Figure 4c**). Both strains also accumulated ~12 mg L⁻¹ of the monoterpene alcohol geraniol, which is commonly produced by yeast with an increased GPP pool (33,36). No farnesol (C₁₅ alcohol) or geranylgeraniol (C₂₀ alcohol) were accumulated by the strains, indicating that subcellular pools of FPP and the C₂₀ geranylgeranyl pyrophosphate (GGPP) were low, and that amplification of limonene synthetic module led to significant redirection of the carbon flux towards monoterpene production.

Q2. I am wondering why the authors chose HPV16 L1 expression, particularly knowing about the insolubility issues reported in bacteria.

Response: HPV16 L1 is the component for the vaccine against HPV infection, which has been previously expressed in S. cerevisiae. The insoluble expression in S. cerevisiae was unexpected; we hoped expressing it in yeast might overcome the problems encountered in bacteria.

Q3. Line 34: the detail of cultivation “on 20 g L⁻¹ glucose” is not necessary as this is standard growing procedure for yeast.

Response: Thanks for reviewer’s suggestion. We have removed ‘on 20 g L⁻¹ glucose’.

Q4. Line 48: Please mention the drawbacks of available methods to clarify the need of this study.

Response: We were a bit confused about this request, since we detailed the drawbacks in the paragraph after the reference to drawbacks in line 48. Perhaps, it was confusing that we referenced drawbacks in one paragraph and then expanded upon those drawbacks in the following paragraph. To avoid confusion, we deleted the sentence at line 48. We have copied the following paragraph here. We think this outlines the drawbacks in good detail but are happy to make any specific improvements suggested:

The brewer’s yeast *Saccharomyces cerevisiae* is a eukaryotic model organism and an important industrial microorganism for production of biofuels, biochemicals, and biopharmaceuticals. In *S. cerevisiae*, multi-copy yeast episomal plasmids (YEPs) or genome integration into ribosomal DNA (rDNA) sites are typically used to increase gene dosage⁵⁻⁸. However, these approaches are not stable in the absence of selection pressure, and plasmids can suffer from copy number instability leading to variable expression levels⁵⁻⁸. In addition, use of selection systems in industrial processes adds additional costs and often is not scalable^{9,10}. To stabilise strains without the need for selective antibiotic or auxotrophy systems, auto-selection markers such as glycolytic genes (*FBA1*, fructose-bisphosphate aldolase; *POT1/TPI1*, triosephosphate isomerase) can be used^{5,11,12}. However, this requires the background strains to have the correct genotype for knock-out. Transposable elements can also be used for multicopy integration, however variable copies are integrated at

random loci on genome, which means integrated components cannot be removed to facilitate future engineering steps (for example, swapping terpenoid synthases for different terpenoid production platforms)¹³⁻¹⁶. A method overcoming all these limitations is highly desirable.

Q5. Line 51: rDNA is not explained

Response: Thanks for pointing out this error. We have revised rDNA into ribosomal DNA (rDNA).

Q6. Line 119: The tested yeast promoters are stated as “data not shown”, however, some of them are presented in supplementary fig. 5 and some were characterized in previous work. So, I consider a list of the characterized promoters would be useful in the supplementary information.

Response: We have now published the “data not shown” work and so we have cited our recently accepted publication, which includes the complimentary promoter-testing work. The sentence now reads as follows:

To identify promoters with suitable expression strengths, promoters were selected from the wide variety of promoters we previously analysed³³, to test with each target locus

Q7. Line 120-126: A supplementary table to describe constructs 1-8 would be useful here, as it is difficult to go back and forth to follow the construct description.

Response: We have noted the constructs 1-8 in Supplementary Table 1.

Q8. Line 171: R is missing in “PBTS1-PL25”. Moreover, the name of the combination promoter/hap gene versus construct number is not consistently used in this chapter and confusing. Both names should be provided for clarity.

Response: Thanks very much for pointing out this mistake. We have corrected this error.

Q9. Line 178: Please specify the genotype of the background strain with an upregulated mevalonate pathway used as it is difficult to identify it in the 3 references.

Response: We have referred to the stain name and the Supplementary Table 2, in which the genotype is described. Please see the revision as below:

We used a background strain with an upregulated mevalonate pathway for production of terpene precursors (o401R, Supplementary Table 2)³⁴⁻³⁶.

Q10. Line 412: Lycopene is not a C30 triterpene, but a tetraterpene (C40) as correctly stated in the abstract.

Response: Thanks very much for pointing out this error. We have corrected all relevant mistakes in the main text.

Q11. The numbering code in figure 6 is not explained.

Response: We have added an explanation in Figure 6 legend, as below:

Numbers in **b** & **c** are for sample cross-reference.

Q12. The methods for LC-MS/MS-based proteomic analysis is missing.

Response: Thanks very much for pointing out the missing. We have added the method in Supplementary Data File 1.

Reviewer #2 (Remarks to the Author):

The authors present a potentially interesting approach for multi-copy integrations and pathway amplifications. There are a few aspects that are lacking in the current manuscript to gain a better appreciation for this proposed approach.

Response: We thank the Reviewer for recognising this work being potentially interesting.

Q13. A very common approach in the field for multi-copy integrations in yeast is via that pITY system developed by Parekh et al from the Wittrup group. This system integrates into the Ty element repeats in yeast and can likewise achieve very high number of stable copies. How does this system reported here compare? A head-to-head experimental comparison of one of the targets (perhaps one of the terpenes) should be included here otherwise the report does not demonstrate novelty.

Response: The HapAmp system is quite different from transposable elements (pITY) mediated multi-copy integration. For pITY system, variable copies are integrated at random loci on genome. This makes recovering selectable marker and removing the integrated components impossible. This is one of the major limitation for pITY system. In the introduction, we describe this limitation, which is shown as below. We have now included a citation to the pITY system in the reference list (reference 17):

Transposable elements can also be used for multicopy integration, however variable copies are integrated at random loci on genome, which means integrated components cannot be removed to facilitate future engineering steps (for example, swapping terpenoid synthases for different terpenoid production platforms)¹³⁻¹⁶.

Q14. Stability is an important factor on multi-copy integrations. The authors must delve into this effect with multiple generations and subcultures to watch for changes in the copy number as a function of long-term culturing.

Response: Thanks for raising for this concern. We have done a stability test by passing down the strain expressing the highest level of GFP for ~48 generation. We found no change in the GFP expression levels after ~48 generation subculturing. We have added the data in Supplementary Figure 4. We added the following description in the main text:

The strain expressed the highest level of yEGFP (Construct 4) was sub-cultured in Yeast extract-peptone-glucose medium for ~48 generations for stability test (Supplementary Figure 4). GFP fluorescence levels and population homogeneity did not change, indicating that HapAmp is genetically stable.

Q15. The authors show a very nice correlation between copy number and expression when it comes to GFP, but this correlation breaks down when looking at most of the products. A further discussion here is warranted.

Response: We added the following discussion in the main text:

“We observed a tight correlation between gene copy number and GFP fluorescence (Figure 2); however, this relationship breaks down for the different terpene products, resulting in variable improvement ratios. This is most likely due to the fact that the relationship between the GFP peptide and its fluorescence is very close and does not rely on other factors such as substrate and cofactor availability – whereas the terpene synthases are enzymes and subject to more influences on their behaviour. In addition, variable metabolic burden caused by overexpression of terpene synthases or other physiological perturbations in metabolically engineered systems may affect the relationship between copy number and product titre. For products, limonene production improvement was ~24-fold...”

Q16. What is the limit of “cargo” size that may be put into this system? How large of a pathway/segment of DNA can be used? More complex pathways should be investigated.

Response: Thanks for raising this concern. In the case for improved lycopene production, we used this system to introduce three expression cassettes, equal to 7917-bp of ‘cargo’. The plasmid size is ~14 kb, which is near to the upper size limitation for pBR322-derivative E. coli plasmids. We have not attempted to determine the limit of ‘cargo’ size. Testing more complex pathways is beyond the scope of the current work. However, we agree that it is worthwhile commenting on the cargo size in the manuscript, and have added the following in the Discussion:

Here we tested several constructs ranging up to three expression cassettes (lycopene pathway: insert size of 7917 bp). We have not sought to test the maximum cargo size for this approach. However, we did not observe a clear relationship between size of the insert (‘cargo’) and copy number amplification, suggesting that even larger inserts may be possible for the technique.

Reviewer #3 (Remarks to the Author):

The manuscript of Peng et al aims to address an important issue in eukaryotic microbial cell factories, that of high level, stable heterologous gene expression. It appears that the capacity of cells to produce enzymes has not been exhausted in yeast. To achieve stability of gene expression and economic feasibility of the production platform, the expression cassettes usually propagated and expressed in plasmids require transfer to the genome in multiple copies. Current approaches to expanding gene copy numbers have been somewhat unsatisfactory.

The proposed new approach takes advantage of haploinsufficiency of certain endogenous genes, to force gene amplification. This is accomplished by substituting the endogenous promoter of the gene with much weaker promoters, which cause the haploinsufficient phenotype. Linking this gene with the expression cassette (flanked by the duplicated terminator sequence and an ARS) causes the simultaneous amplification of the cassette to compensate for the growth defects caused by low gene expression levels. This approach was tested with two haploinsufficient genes and eight different weak promoter constructs. Amplified copies varied between 5-47 copies and GFP reporter levels reaching in the highest case 92-fold over single copy integrants. The system was validated with three different terpene synthases and a viral protein.

Overall, it is a well written manuscript that has the potential to move the field of Microbial Cell Factories closer to expanded industrial utilization.

Response: We thank the Review for recognising the significance of this work and appreciate the comments for improvement.

Questions:

Q17. 1. In different constructs, the ARS used are different. Is the nature of the ARS expected to play a role in cassette amplification?

Response: Thanks for raising this concern. Previous literatures indicate ARS was important for ribosomal Amplification (DOI 10.1016/j.molcel.2009.07.012), amplification of chorion (eggshell) gene in Drosophila ovarian follicle cells (DOI: 10.1101/gad.822101), and oncogene amplification (DOI: 10.1002/gcc.20448). To address the Reviewer's concern, we have tested a HapAmp construct with ARS removed. Without ARS, HapAmp can happen, shown by the formation of the bright colonies on the transformation plate. To revise the manuscript, we added the relevant result in Supplementary Figure 1 and the main text as below:

In the initial design (Figure 1), we include ARS in the module basing on the genetic features at naturally amplified genomic loci. To confirm the role of ARS in the current system, we removed the ARS sequence in the Construct 3. The ARS-removed construct could lead to the formation of the very fluorescent colonies after transformation (Supplementary Figure 1). This indicates that ARS may not be essential for HapAmp.

Q18. 2. Could the authors test levels of expression of the two haploinsufficient genes subsequent to amplification and compare them with pre-amplification levels, as well as the wild type ones?

Response: We did not test the levels of the expression levels of the haploinsufficient gene in wild-type and HapAmp strains. It could be interesting, but not necessary, for this supplementary confirmation of the HapAmp mechanism. Unfortunately, we could not coordinate this experiment, because of the limit on experimental resources for further running this work.

Q19. 3. Figure 3b It would be helpful to the readers to annotate the region of amplification on the figure, and include the far-right T sequence. The same for figure 4a and 5a.

Response: We have modified the Figure 3b, Figure 4a, Figure 5a, and Figure 6a by annotating the amplified region and the far-right RPL25 terminator.

Q20. 4. L395 post-diauxic-shift. Is this a mistake?

Response: Thanks for picking up this mistake. We have corrected this mistake, which was in the legend of supplementary figure 6.

Reviewers' Comments:

Reviewer #1:

Remarks to the Author:

Peng et al. addressed all my comments in the revised version of this manuscript. I consider the resubmitted draft as excellent and the added experiments highly appropriate. Thus, I have no suggestions for additional changes and recommend the acceptance of the manuscript.

Reviewer #2:

Remarks to the Author:

The authors have sufficiently updated their manuscript and responded to reviewer comments. I would recommend publication as is.

Reviewer #3:

Remarks to the Author:

The authors' response and changes implemented are satisfactory